# The Need for Nigeria to Embrace the Hygiene Rating Scheme

**Hope Akegbe** [1,*]**, Helen Onyeaka** [2,*]**, Adeola Dolapo Omotosho** [3]**, Chidinma Ezinne Ochulor** [4]**,
Esther Ibe Njoagwuani** [1]**, Ifeanyi Michael Mazi** [1]**, Iyiola Olatunji Oladunjoye** [5]**, Ogueri Nwaiwu** [6]**,
Olumide A. Odeyemi** [7] **and Phemelo Tamasiga** [8]

1    Department of Microbiology, Faculty of Life Sciences, University of Benin, Benin City 300213, Nigeria
2    School of Chemical Engineering, University of Birmingham, Edgbaston, Birmingham B15 2TT, UK
3    Department of Cell Biology, Faculty of Life Sciences, University of Lagos, Lagos 101017, Nigeria
4    Department of Food Science and Technology, Faculty of Agriculture, University of Nigeria,
     Nsukka 410001, Nigeria
5    Department of Microbiology, Faculty of Life Sciences, University of Ilorin, Ilorin 240003, Nigeria
6    School of Biosciences, University of Nottingham, Sutton Bonington Campus, Loughborough LW12 5RD, UK
7    Research and Research Training Portfolio, Academic Division, University of Tasmania, Hobart 7005, Australia
8    Public Policy in Africa Initiative, Yaounde, Cameroon
*    Correspondence: hopeakegbe9@gmail.com (H.A.); h.onyeaka@bham.ac.uk (H.O.)

**Abstract:** Foodborne diseases pose a primary global health concern, affecting people across high- and low-income countries, with the less privileged often suffering the most. This research proposes the adoption of a Hygiene Rating Scheme (HRS) to help customers make informed decisions about where and what to eat. The scheme has already demonstrated success in countries such as the United States, Northern Ireland, Wales, and England in reducing the risk of foodborne diseases. This article highlights the significance of Nigeria embracing the HRS and its potential to combat foodborne diseases. Adopting the scheme will incentivize food business owners to improve their sanitary conditions and food quality by implementing Good Manufacturing Practices (GMPs). The scheme's transparent inspection results make it easier for customers to choose higher-rated outlets, reducing the cost of disease outbreaks and promoting public health. In conclusion, the HRS provides a practical solution to addressing the issue of foodborne diseases and promoting food safety.

**Keywords:** hygiene rating scheme; foodborne disease; food safety; Nigeria

## 1. Introduction

Foodborne diseases are global health threats, with one in ten illnesses caused by contaminated food [1]. The World Health Organization (WHO) estimates that over 200 diseases are linked to food consumption containing microorganisms or chemical substances such as heavy metals [2], particularly in Africa and Southeast Asia [3]. The United Kingdom's (UK) Food Standards Agency (FSA) oversees food safety by providing information and guidance [4].

The Food Hygiene Rating Scheme (FHRS) was rolled out gradually in England, Wales, and Northern Ireland in November 2010 [1]. The Food Hygiene Information Scheme (FHIS), a similar program, began full deployment in January 2009 after being trialed in Scotland from November 2006 to January 2009 [5]. The FHRS is run by the Food Standard Agency (FSA) in partnership with local authorities; however, the system differs among the three countries. The FHRS uses a hygiene rating between 0–5 to grade the food quality. On the other hand, the Food Standard Scotland (FSS) runs the FHIS and employs the word hygiene rating (pass or improvement required) [5]. The FHRS and FHIS inform the public about the food hygiene standards of food businesses so they can make informed decisions about where to eat or purchase food [2]. The standards set by these schemes have driven up food hygiene standards in food businesses that aimed to meet consumer demand and have incorporated them into their system [6]. The FHRS in the UK covers businesses that

sell food directly to customers, including grocery stores and food establishments such as restaurants, takeaways, sandwich shops, cafés, and other places where food is prepared [3]. Some food-selling businesses are excluded from grading due to specific rules [7]. In the UK, food businesses must implement and maintain Hazard Analysis and Critical Control Point (HACCP)-based hygiene practices to receive a hygiene rating after full or partial inspection. Since food business operators are required to display the result of their hygiene inspection, in England, Wales, Northern Ireland, and Scotland, they are motivated to maintain or improve hygiene standards [5].

Despite the success of FHRS in countries such as the UK, a lot needs to be done to implement food safety rating systems in Nigeria, where foodborne diseases have a significant impact. This paper evaluates the current state of Nigeria's food hygiene using reliable, valid, and up-to-date secondary data, discusses the FHRS and its potential impact, highlights the challenges, and proposes measures for its successful implementation in Nigeria. Secondary data used in this study were collected from peer-reviewed journal articles, reports, conference papers, and internet articles. Keywords such as *Hygiene rating scheme*, *foodborne disease*, *food safety*, and *Nigeria*, were entered into search engines such as "Google Scholar", "Scopus", "Web of Science", and "PubMed". Several materials were generated during this study; however, only 57 were put forward for use. They comprise 29 recent journal articles, 14 recent studies published online, 1 textbook, and 13 reports, all relevant to the topic of discussion. All materials used were collected using search engines such as "Google Scholar", "Scopus", "Web of Science", "PubMed", and "Google". The materials were screened based on some inclusion and exclusion criteria, such as relevance to the study objective and the quality and recency of the source material. However, some reports, such as the FAO (2005), were still used since they contained vital information and had not been revised. Appendix A contain more information on data collection.

## 2. Current State of Food Hygiene in Africa

Approximately 600 million individuals globally experience food hygiene-related illnesses after consuming food, and an additional 420,000 die from food hygiene-related complications each year [8]. Food hygiene is the absence of harmful biological, chemical, or physical agents in food [9]. Poor food hygiene practices can result in foodborne diseases, which can cause severe and long-lasting health effects [9].

A study by WaterAid in southern African countries (Lesotho, Malawi, Zambia, Zimbabwe, Madagascar, Swaziland, and South Africa) found nearly 70% of diarrhea in developing nations is caused by pathogens from food [10]. The study also noted that these countries lack food hygiene data compared to sanitation, although some hygiene policies are included in the education policies of South Africa, Lesotho, and Malawi. The other four countries have food hygiene awareness in their strategic nutrition policies [10].

The National Agency for Food and Drug Administration and Control (NAFDAC) in Nigeria regulates pathogens and chemicals in processed foods but not in abattoir products, fish, agricultural products, and fruits and vegetables. The AfricaSan movement, led by African Ministers Council on Water (AMCOW), aimed to bridge hygiene gaps in Africa and held a conference in Kigali, Rwanda, in 2022 to build political support for hygiene and sanitation [11].

Ensuring food hygiene and safety requires qualified inspectors and proper inspection procedures. However, in Nigeria, food inspection faces challenges such as limited personnel with varying abilities, inadequate logistics support, and heavy workloads within a limited timeframe. National food assessment services are primarily concentrated in urban areas, limiting access to rural regions and small towns where food production and farming are prevalent [12].

Despite the importance of health and nutrition, limited data exists on food hygiene. For example, the UNICEF WASH Humanitarian Action (HAC) for children in 2022 focuses on water, sanitation, and hygiene in emergencies but does not address food hygiene specifically.

The HAC aims to promote hygiene and reach over 35 million people in African and Asian countries [13].

In many countries, processed foods are considered more hygienic due to monitoring and testing based on GMPs, Good Hygiene Practices (GHPs), Sanitation Standard Operating Procedures (SSOPs), and HACCPs [14]. However, in most African countries, excluding South Africa and Egypt, fast food from restaurants and eateries must be regularly checked for hygiene. Street food is the least regulated for hygiene. Lagos, Nigeria, conducts routine checks on restaurants but needs more initiatives.

In Nigeria, the lack of competent authorities for protecting consumers from the dangers of food consumption, providing information, and effectively communicating food hygiene to food business operators and consumers results in a lack of a proper food hygiene system [12]. Nigeria must implement an inspection scheme, establish laboratory support services, adopt a food hygiene and standard plan, and develop a consumer education framework to address this.

*Current State of Food Hygiene in Nigeria*

The current state of food hygiene in Nigeria is a significant concern due to the high incidence of foodborne illnesses resulting from poor hygiene practices in the food industry [15]. Nigeria, a developing country, has a high burden of foodborne illnesses that contribute to morbidity and mortality in the population [16]. This problem is exacerbated by inadequate infrastructure, a need for proper regulations, and limited awareness of food hygiene practices among food handlers and consumers [16].

Most food handlers in Nigeria lack formal education and training on food hygiene practices, a significant contributing factor to the high prevalence of foodborne illnesses. Additionally, most food sold in open markets in Nigeria is contaminated with various microorganisms, including bacteria and fungi [17,18].

The Nigerian government has tried to address the issue of food hygiene by enacting regulations such as the NAFDAC FHR [9]. However, these regulations need to be more effectively enforced, and there is a need for more monitoring and surveillance mechanisms to ensure compliance. A report from an extensive study indicates that of the existing legislation relating to food safety, only 14 out of 16 (87.5%) were enacted over a decade ago, while a lot are overdue for review or repeal. Nigeria's Food Safety and Quality Bill, produced in 2016, still needs to be passed into law (Act) at the National Assembly [19,20]. The above situations clearly describe Nigeria's incapacitated food hygiene and safety state.

There is a need for increased awareness campaigns on food hygiene practices among food business operators, handlers, and consumers. In addition, there is a need for the government to strengthen regulations and enforcement mechanisms to ensure compliance with food hygiene standards. This study introduces a simple and much easier-to-achieve concept that can improve Nigeria's food safety standards with minimum cost.

## 3. Hygiene Rating Scheme (HRS)—How it Works

The HRS guides consumers by displaying businesses' hygiene standards with a rating from 0 to 5, in-store and online, allowing for informed food choices [21].

The Figure 1 below shows the level of hygiene standards; there are six varying food hygiene ratings.

Independent research findings with consumers led to the use of numbers and simple word descriptors for rating as seen in Figure 1 [22].

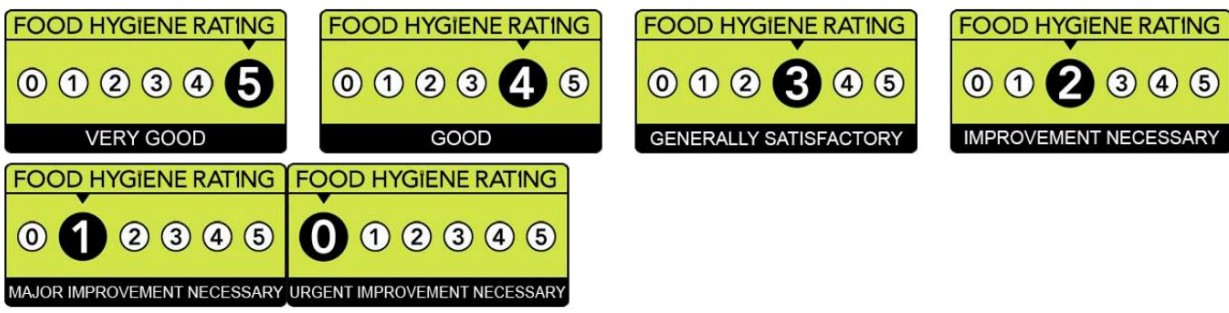

**Figure 1.** The rating scale.

The Food Law Code of Practice outlines critical elements for calculating the FHR in an audit, examination, or partial evaluation. These elements include the following:

- The degree of safety procedures and food hygiene compliance, including food-handling practices and temperature control.
- Degree of current adherence to structural requirements, including the structure's state, cleanliness, lighting, ventilation, and layout.
- Assurance in management/control procedures [22].

The level of compliance with the elements above determines the food hygiene rating. An establishment's FHR depends on the food hygiene intervention rating following an inspection, partial inspection, or audit by a food safety officer [22]. However, rating systems show variation internationally, from symbols (smiley faces or starts), letter ratings (e.g., 'A', 'B'), numerical scores, or statement cards (e.g., pass, closed) [23].

PLUS

Table 1 shows the Food Hygiene Intervention Rating Guide [24]. When a considerable risk of *Clostridium botulinum* contamination is detected in food, add a score of 20 to the previous score—the microorganism's survival or multiplication during processing, or pathogenic microorganisms or their toxins contaminating ready-to-eat food, e.g., *Salmonella* sp.; *Bacillus cereus*, *E. coli* 0157 or other VTEC. Table 2 below shows the conversion of intervention-rating scheme scores to food hygiene ratings [25].

**Table 1.** Food Hygiene Intervention Rating Guide.

| | Scoring System Guide |
|---|---|
| 30 | Unsatisfactory conformity track record. Need for food safety knowledge and awareness. Need for an adequate understanding of potential risks and quality management. Food safety control procedures are missing. Adequate recognition of the significance of food safety and proper hygiene control techniques is needed. |
| 20 | There is a significant disparity in the compliance record. Food safety understanding and expertise still need to be improved. Infraction of hazards and mitigation systems. Following the latter evaluation of food hygiene conditions, food safety control procedures and improvements needed to be included. Some people need to be more open to recognizing the importance of food safety and hygiene control procedures. |
| 10 | The conformity record is substantial. The existence of critical food safety guidance sources and GMP guides corresponds to the nature of the business. Recognition and comprehension of pertinent hazards and preventive strategies. Satisfactory integration and execution of food safety management processes based on well-documented procedures. Officials will ensure the company is making significant progress traceable to food safety management procedures. Suppose the initial non-compliances have been acknowledged and dealt with. In that case, however, new non-compliances have surfaced, and cumulative hazards have remained the same; a score of 10 will be assigned for more than one intervention cycle. |

**Table 1.** *Cont.*

| | Scoring System Guide |
|---|---|
| 5 | A satisfactory history of compliance. The availability of food safety advice in-house or technical advice from a Primary or Home Authority is made accessible and put into use. Managing hazards control effectively. Effectively employing self-checks with well-documented food safety management procedures and aligning with the business type. Confirm general procedure compliance through an audit by a competent authority, where relatively insignificant non-compliances are not identified as critical to food safety. |
| 0 | Outstanding conformity record. Food safety advice is readily accessible in-house, or technical guidance from a local authority is available and used. The Food Business Operator demonstrates expertise and knowledge. Demonstrate effective self-checks with detailed records of food safety monitoring processes that are appropriate for the business type and may include external audit systems. The competent authority's audit confirms excellent adherence to food safety procedures. |

**Table 2.** Chart showing the conversion of intervention-rating scheme scores to food hygiene ratings.

| Tracing numerical scores from the Food Law Code of Practice intervention-rating framework to the six FHRS food hygiene evaluations. | | | | | | |
|---|---|---|---|---|---|---|
| Scores for total intervention rating | 0–15 | 20 | 25–30 | 35–40 | 45–50 | >50 |
| Other criteria for scoring | There is no independent score higher than 5 | There is no independent score higher than 10 | There is no independent score higher than 10 | There is no independent score higher than 15 | There is no independent score higher than 20 | - |
| Food hygiene rating | 5 | 4 | 3 | 2 | 1 | 0 |
| Signifier | Very good | Good | Generally satisfactory | Improvement is required | Significant improvement is required | Urgent improvement is required |

## 4. Adoption and Implementation of the Hygiene Rating Scheme in the UK

The UK government introduced the FHRS to improve food hygiene standards, promote transparency, and empower consumers to make informed choices about where to eat. The scheme was first launched in 2010 and has since been widely adopted nationwide [26].

The UK government worked closely with local authorities, food safety agencies, and industry stakeholders to develop and implement the HRS [27]. Extensive consultations were conducted to gather feedback and insights from various parties. This collaborative approach ensured that the scheme was feasible, practical, and well-received by the industry [28]. The implementation of the scheme involved several key steps:

### 4.1. Standardized Inspection Criteria

A set of criteria and guidelines were established to assess food establishments' hygiene practices [29]. These criteria cover food handling, cleanliness, and management procedures.

### 4.2. Inspection and Rating Process

Trained environmental health officers to conduct routine inspections of food establishments. They assess compliance with the established criteria and assign a hygiene rating on a scale of 0 to 5. A rating of 5 indicates excellent hygiene practices, while 0 signifies that urgent improvement is required [29].

*4.3. Displaying Ratings*

Food businesses were mandated to display their hygiene ratings on their premises prominently. This enabled consumers to easily view and consider the hygiene ratings when making dining decisions [30].

*4.4. Online Accessibility*

The hygiene ratings are also published on a dedicated website, allowing consumers to search and compare ratings for different establishments [31]. This online accessibility further enhances transparency and encourages businesses to maintain high hygiene standards.

The adoption of the HRS in the UK has yielded several significant impacts and benefits.

*4.5. Improved Food Safety*

The scheme has driven food establishments to prioritize and enhance their hygiene practices. Businesses strive to achieve higher ratings, leading to improved food safety standards across the industry [32].

*4.6. Consumer Empowerment*

The hygiene ratings empower consumers by providing them with easily accessible information about food establishments' hygiene standards. This transparency enables consumers to make informed choices and dines at higher-rated places, encouraging businesses to maintain good hygiene practices [30].

*4.7. Industry Reputation and Compliance*

The scheme has positively influenced the overall reputation of the food service industry in the UK. Businesses with higher ratings are perceived as more trustworthy, leading to increased customer confidence. Moreover, the scheme has encouraged compliance as establishments strive to obtain better ratings [33].

*4.8. Continuous Improvement*

The scheme operates as a dynamic process, prompting businesses to improve their hygiene practices continually. The regular inspections and public display of ratings create a feedback loop, motivating establishments to address any deficiencies and maintain high standards [34].

Adopting and implementing the HRS in the UK also faced certain challenges, including initial resistance from some food business owners, inconsistencies in interpretation by inspectors, and resource constraints for inspections [30]. However, ongoing efforts are made to address these challenges by providing support, training, and clear guidelines to inspectors and businesses.

## 5. The Need for Nigeria to Embrace the Hygiene Rating Scheme

Nigeria needs to adopt the HRS to ensure food safety, which directly impacts public health [35]. Implementing a credible evaluation tool such as the FHRS can help address Nigeria's primary challenge of needing more expertise and information in investigating foodborne disease outbreaks. The FHRS can facilitate health surveillance and monitoring, prevent foodborne disease outbreaks, and promote a food safety culture among food service entrepreneurs.

Adopting this framework has the potential to improve food safety in Nigeria and deliver various benefits such as the following:

*5.1. Effective Risk Communication with Customers*

Consumers will have access to transparent information about the hygiene standards of food outlets, enabling them to make informed decisions about where to shop for food and dine.

The scheme rating on in-store and online hygiene standards, accessible to consumers, allows them to make informed choices about the food they consume and where they purchase it.

This information empowers consumers to prioritize their health and hold food service entrepreneurs accountable for maintaining high hygiene standards. This creates a positive feedback loop, where food outlets with high hygiene ratings are rewarded with increased business, and those with lower ratings are incentivized to improve their standard.

### 5.2. Improve Global Perception of Nigeria's Food Sector

Adopting FHRS provides a standardized and transparent approach to evaluating hygiene standards in Nigeria's food sector. This can enhance Nigeria's reputation as a trustworthy producer and supplier of safe, high-quality food, dispelling negative perceptions and attracting investment and support. By establishing a credible and effective food safety framework, Nigeria can improve its economy and help to attract investment and support in the food industry, contributing to the overall development of Nigeria.

### 5.3. Reduced Risk of Foodborne Illnesses

The FHRS aims to prevent foodborne illnesses by promoting high hygiene standards and holding food service entrepreneurs accountable for implementing GMP. Providing explicit information about hygiene standards helps customers make informed decisions, reducing exposure to contaminated food. Encouraging effective food safety measures through accountability minimizes the risk of foodborne diseases, resulting in a safer food environment and a reduced burden of illnesses in the community.

### 5.4. Reduction in Crucial Violations Associated with Foodborne Diseases and Increased Adherence to Food Safety Regulations

Evaluation of food business operators in Nigeria using the HRS would likely improve general compliance with food safety regulations, as similar studies in Scotland showed a positive result [6]. Encouraging businesses to adopt improved hygiene standards, the FHRS helps reduce food poisoning occurrences. The scheme's significance should be considered, as its effect is evident in regions such as England, Wales, and Northern Ireland, where 96% of food enterprises currently have a rating of three ('usually satisfactory') or above [36]. Further studies have revealed that food establishments with higher ratings are usually not implicated in foodborne disease outbreaks. According to the FAO and reports from research studies, the leading causes of foodborne outbreaks are suboptimal food holding and storage conditions and poor hygiene practices [37,38]. In a research survey, the authors discovered that following a preliminary food hygiene rating assessment, food services became more conscious of how food was handled and stored [5].

Additionally, they included a food safety officer who ensured the implementation and strict adherence to food safety measures, proper documentation, and accurate tracking of records [5]. If effectively implemented, the food hygiene rating program would improve food hygiene competitiveness among companies, boosting the number of facilities that exceed food hygiene regulatory criteria. Higher food hygiene standards, as observed in the UK, would abate the prevalence of foodborne diseases among the Nigerian population [5].

### 5.5. Promotion of Healthy Competition among Food Service Owners

A grading and inspection score system would encourage "healthy competition" among food industry operators [39]. Owners of Nigerian food enterprises would likely invest in food safety concerns to earn a better grade, minimizing the danger of foodborne infections. Restaurant management would be incentivized to adhere to GMP and HACCP standards if they were aware that their reputation and rate of patronization were reliant on the grade they received. This will enhance general standards in food businesses, which may increase income for food business owners. It will encourage them to adhere to food hygiene and safety regulations. They will gain more clients, making them more profitable as a result.

## 6. Challenges of Implementing Hygiene Rating Scheme in Nigeria

**Lack of resources:** Nigeria needs more resources for public health and food safety, which may make implementing and sustaining the scheme challenging.

**Limited infrastructure:** In some parts of Nigeria, there may be limited infrastructure and technical expertise to support the scheme's implementation, including a lack of trained food safety inspectors, inadequate laboratory facilities, and insufficient data management systems.

**Cultural and language barriers:** In a culturally diverse country such as Nigeria, language and cultural barriers may make it difficult to effectively communicate the objectives and benefits of the scheme to the public.

**Resistance from food service entrepreneurs:** Some may resist the scheme due to concerns about the cost and time required to implement it and potential adverse effects on their business.

**Inadequate legal framework:** In some countries, the legal framework for food safety may need to be revised, making it difficult to enforce the HRS and ensure that food service entrepreneurs comply with the requirements.

**Limited consumer awareness:** In some areas, consumer awareness about the importance of food safety may be limited, making it challenging to engage the public in supporting the scheme.

Wales, England, Northern Ireland, Scotland, and India are countries where the FHRS is utilized in food establishments. In such countries, FHRS promotes food businesses where hygiene is prioritized as consumers are naturally drawn to eating in top-rated food outlets against those with poor ratings [38].

Implementing a HRS in Nigeria will be a step in the right direction toward reducing the incidence of foodborne diseases. This is because consumers will have the correct information to influence their decision when purchasing food. However, implementing a profound change such as this will bring out challenges based on how key players react. People react differently to change depending on how favorable or unfavorable the change is to them. The most common reactions to change are adaptation and resistance. In their theories of change, Husain and Morris described how implementing the FHRS brings change that cuts across food consumers, regulatory bodies, and food businesses.

Therefore, identifying the challenges of implementing an HRS requires looking through the lens of the key players—the consumers, regulatory bodies, and food businesses [40].

### 6.1. Consumers

Successful implementation of the FHRS depends on how well consumers utilize the new information.

The premise is that information about the FHRS of food businesses will be accessible to all consumers. Poor circulation of information, especially in rural areas, has been identified as a significant hindrance to food security in Katsina state, Nigeria [41]. Food safety is an inseparable aspect of food security. Additionally, poor internet connection and the high cost of internet data bundles can hinder consumers from accessing information about the FHRS from the website [40].

In Nigeria, fast food restaurants have higher hygiene compliance than street food sellers. However, street food sellers continue to garner high patronage due to the low cost of their food in contrast with fast food restaurants. A survey conducted in Nigeria to understand why residents patronize street food sellers despite the food safety implications showed that the low food cost resonated with all respondents. Fast food restaurants stand a better chance of scoring high if the FHRS is adopted [42]. High scores may translate to further price hikes making food even less affordable for low-income earners who would have no choice but to continue patronizing low-rated food businesses that offer affordability [38].

*6.2. Regulatory Bodies*

Successful implementation of the FHRS in Nigeria depends on regulatory bodies' willingness and ability to take on the new responsibility. Several organizations in Nigeria are tasked with ensuring the safety and quality of food products. These entities include the NAFDAC, which regulates the manufacture, distribution, importation, and advertising of food, drugs, cosmetics, medical devices, and chemicals in Nigeria [43]. The Standards Organization of Nigeria (SON) develops, publishes, and enforces quality standards for all products, including food [44]. The Federal Ministry of Agriculture and Rural Development (FMARD) promotes agricultural development and food security in Nigeria. The Consumer Protection Council (CPC) protects consumer rights by preventing unfair trade practices and enforcing product safety and quality standards. The National Agricultural Extension and Research Liaison Services (NAERLS) conducts research on agricultural practices, disseminates information to farmers, and promotes food safety and security [45]. Finally, the Nigerian Institute of Food Science and Technology (NIFST) is a professional body that promotes food science and technology, ensuring that food products are safe and of high quality. Nigeria possesses these regulatory bodies, but the inspection processes need to follow a risk-based approach in their inspection investigations [46]. Most laboratories need to be better equipped, risk analysis tools need to be improved, and there needs to be more training for inspectors [47].

*6.3. Food Businesses*

Implementing the FHRS in Nigeria is expected to motivate food businesses to improve their hygienic standards as consumers will be making hygiene-rating-influenced food purchases [48].

However, to survive Nigeria's tough economy, businesses may transfer hygiene expenses (e.g., labor, sanitizers, water, stations) to consumers via food price hikes. Smaller food companies unable to compete may go bankrupt, worsening unemployment. High food prices restrict households' ability to pay for healthcare and education. Research on the effects of food price hikes on families showed that spending half of their earnings on food limits their ability to afford other essentials such as healthcare and education [49].

## 7. Way Forward

### 7.1. Proposed Measures to Help Realize Hygiene Rating Scheme in Nigeria

These measures can help realize the implementation of the HRS in Nigeria:

**Building capacity:** Building capacity in areas such as food safety inspection, laboratory testing, data management, and communication will be essential to implementing the scheme successfully.

**Involving stakeholders:** Involving stakeholders such as government agencies, food service entrepreneurs, and consumer organizations in the scheme's development and implementation can help ensure its success.

### Government Agencies and Local Authorities

To adopt the FHRS in Nigeria, NAFDAC must be actively involved. NAFDAC would have to collaborate with other private food safety agencies in Nigeria such as Rentokil Boecker to devise measures to create awareness of the scheme among food business operators, conduct hygiene inspections, upload results of food business ratings to their website, and if possible, mandate food business operators to display the result of ratings. They must provide training and support to ensure the effective operation of the scheme. They must also be willing to take the right actions against food businesses with very low hygiene ratings. The state and federal governments must provide sufficient funding and resources for the smooth running of the scheme.

### Food Business Operators

Food business operators must be willing to engage in hygiene inspections and comply with regulators to display results of ratings when mandatory.

**Establishing legal framework:** Establishing a robust legal framework for food safety, including laws and regulations that support the implementation of the HRS, will be important in ensuring that food service entrepreneurs comply with the requirements.

**Consumers' awareness-raising:** Raising awareness about the importance of food safety and the benefits of the HRS among the public can help to encourage their support and engagement. Once implemented, consumers should assess hygiene information about food business premises through FHRS 343 certificates and FSA websites.

**Partnerships:** Building partnerships between government agencies, public health organizations, food service entrepreneurs, and other relevant stakeholders can help to ensure that resources and expertise are leveraged to support the implementation of the scheme.

**Incentives:** Providing incentives such as tax breaks, technical assistance, and marketing support to food service entrepreneurs who comply with the requirements of the HRS can encourage their participation and support.

**Monitoring and evaluation:** Regular monitoring and evaluation of the scheme can help to identify areas for improvement and ensure its continued success in reducing the risk of foodborne illnesses in Nigeria.

To successfully implement an HRS in Nigeria, food safety training and capacity building are crucial, which should be carried out by government institutions responsible for public food safety. Exercise can help close the knowledge gap among food vendors regarding food safety, hygiene, and sanitation and emphasize the importance of having a rating scheme to assess food handlers and vendors in Nigeria. A study in Iwu, Nigeria, found that 63% of food vendors sampled needed to follow good hygiene practices due to a lack of food safety training and awareness [50]. The government should mandate enforcing food safety control processes such as HACCP, good manufacturing practices (GMPs), and ISO accreditations for small food businesses in Nigeria. This can be achieved by delegating food regulation to local government units to oversee street vendors. South Africa's ISO/IEC 17020 veterinary service accreditation can serve as a model for Nigeria [50]. The quality control ensured by the implementation of these food safety management systems in ensuring meat safety for the general populace in South Africa has led to the establishment of the National Abattoir Hygiene Rating Scheme (NAHRS) Committee, which oversees the inspection of activities within abattoirs to ensure the distribution of safe and hygienic food [50]. Several kinds of research have shown the public health risks and implications of abattoirs that lack good hygienic practices and food safety standards in various African countries [51–54]. Therefore, implementing proper food safety management systems across wet markets would foster HRS cultivation and regular monitoring and enforcement by government stakeholders.

Food safety policies in Nigeria should be improved to include standardized HRS in ensuring food safety within the region. The inclusion of these HRS should be communicated amongst relevant food stakeholders. Furthermore, this should be included in food safety audits and checks carried out in food manufacturing and services companies to ensure proper implementation. Food safety researchers should conduct studies evaluating the delivery and impact of implementing food HRS in Nigeria [5].

Finally, to increase support for implementing food HRS in Nigerian food production and manufacturing industries, food safety governmental institutions should establish databases where the hygiene ratings of these companies can be publicly disclosed at designated intervals after regular inspections and audits [55].

*7.2. Recommendations*

Based on the benefits and challenges discussed, the following recommendations can be made to help realize the implementation of the HRS in Nigeria:

**Increase funding for food safety:** Increased funding for food safety initiatives can help build the capacity of inspection and laboratory systems and support the implementation of the HRS.

**Engage the private sector:** The private sector should be engaged in developing and implementing the scheme to ensure that its expertise and resources are leveraged to support its success.

**Develop a solid legal framework:** A robust legal framework that supports the implementation of the HRS should be developed and enforced to ensure that food service entrepreneurs comply with the requirements.

**Raise awareness:** Awareness-raising campaigns should educate the public about the importance of food safety and the benefits of the HRS, encouraging their support and engagement.

**Foster partnerships:** Partnerships between government agencies, public health organizations, food service entrepreneurs, and other relevant stakeholders should be fostered to support the implementation and success of the scheme.

**Provide incentives:** Incentives should be provided to food service entrepreneurs who comply with the requirements of the HRS, encouraging their participation and support.

**Monitor and evaluate:** Regular monitoring and evaluation of the scheme should be conducted to identify areas for improvement and ensure its continued success in reducing the risk of foodborne illnesses in Nigeria.

## 8. Conclusions

In conclusion, Nigeria could significantly enhance food safety and decrease foodborne illnesses by adopting the HRS. Consumers can access food outlet sanitation data, enabling them to make informed decisions about where and what to eat. The scheme motivates food entrepreneurs to adopt GMPs, promoting food safety. Although implementing the scheme in Nigeria is difficult, stakeholders can guarantee its success by building capacity, creating a legal framework, raising awareness, establishing partnerships, providing incentives, and monitoring and assessing its impact. Stakeholders must collaborate to guarantee the scheme's successful implementation and long-term improvement to realize its advantages fully.

**Author Contributions:** H.A. developed the concept for this review. H.A., A.D.O., C.E.O., E.I.N., I.M.M. and I.O.O. wrote the first draft of the manuscript. H.O., O.A.O., P.T. and O.N. proofread and edited the language. H.A. and H.O. revised the manuscript. H.O. supervised the project and critically revised the final manuscript. All authors contributed to the manuscript and approved the submitted version. All authors have read and agreed to the published version of the manuscript.

**Funding:** This research received no external funding.

**Institutional Review Board Statement:** Not applicable.

**Informed Consent Statement:** Not applicable.

**Data Availability Statement:** Not applicable.

**Acknowledgments:** The authors appreciate those who have contributed to the success of this article.

**Conflicts of Interest:** The authors declare no conflict of interest.

## Appendix A  Bibliographic Review

### Appendix A.1 Introduction

This section provides a bibliographic review of the current state of Nigeria's food hygiene, focusing on the evaluation of the Food Hygiene Rating Scheme (FHRS), its potential impact, challenges, and proposed measures for successful implementation. The review utilizes reliable, valid, and up-to-date secondary data obtained from various sources, including peer-reviewed journal articles, reports, conference papers, and internet articles.

### Appendix A.2 Methods

To gather relevant secondary data, a comprehensive search strategy was employed. Keywords such as "Hygiene rating scheme," "foodborne disease," "food safety," and

"Nigeria" were used to search databases and search engines including "Google Scholar," "Scopus," "Web of Science," and "PubMed."

*Appendix A.3 Data Collection and Selection*

The search yielded numerous materials related to the topic. However, a rigorous screening process was conducted to select the most suitable sources for this study. Inclusion and exclusion criteria were applied, considering factors such as relevance to the study objective, quality, and recency of the source material.

*Appendix A.4 Inclusion Criteria*

Relevance to the study objective: Materials and sources that directly address or provide valuable insights into the study were considered.

Quality of the source material: Only materials from reputable and reliable sources such as peer-reviewed journal articles, reports from reputable organizations, conference papers, and relevant textbooks were considered.

Recency of the source material: Only materials published within the last ten years were considered, except for exceptional cases where the material is highly relevant.

Connection to Nigeria: Only materials focusing on or including information specifically relevant to Nigeria's food hygiene landscape, policies, practices, or challenges.

Availability of full text: Only materials with accessible full-text versions written in English were considered evaluation and analysis.

*Appendix A.5 Exclusion Criteria*

Irrelevance to the study objective: Sources that did not directly contribute to the evaluation of Nigeria's food hygiene, FHRS, challenges, and proposed measures were excluded.

Poor quality or lack of credibility: Sources that lacked rigorous research methodologies, contained significant biases, or had questionable credibility were excluded.

Outdated or obsolete information: Sources that were outdated (2012 and below) and have been superseded by more recent studies were excluded, except for cases where the information is considered crucial and has not been revised.

Lack of connection to Nigeria: Sources that did not address Nigeria's food hygiene situation or lacked specific relevance to the country's context were excluded.

Inaccessibility: Sources that were unavailable or have restricted access or were written in languages other than English, making them inaccessible for proper evaluation and analysis were excluded.

*Appendix A.6 Results*

Out of the materials generated, a total of 57 sources were deemed appropriate for inclusion in this bibliographic review. These sources consist of 29 recent journal articles, 14 recent studies published online, 1 textbook, and 13 reports. Each of these sources contributes relevant information pertaining to Nigeria's food hygiene.

Despite the emphasis on recent sources, some reports, such as the FAO (2005), were still considered due to their significance and the absence of subsequent revisions.

*Appendix A.7 Discussion*

The selected sources provide valuable insights into the current state of Nigeria's food hygiene, with a specific focus on the FHRS. These materials facilitate a comprehensive analysis of the scheme's potential impact, shed light on existing challenges, and offer proposed measures for successful implementation in Nigeria.

*Appendix A.8 Conclusions*

In conclusion, this bibliographic review synthesizes reliable, valid, and up-to-date secondary data to evaluate Nigeria's food hygiene. The review highlights the FHRS, its

potential impact, challenges, and proposed measures for successful implementation. The inclusion and exclusion criteria applied during the selection process ensured that the sources chosen contribute to the overall understanding of the topic. By drawing from a diverse range of materials, this review provides a comprehensive overview of Nigeria's food hygiene landscape.

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
