# Peer review of "The Need for Nigeria to Embrace the Hygiene Rating Scheme"

_2673-947X, doi:10.3390/hygiene3020016_

Round 1

Reviewer 1 Report

Dear Authors 

Many thanks for the oppourtunity to review your manuscript. 

While i see the importance of the work however in my opinion the paper will need further attention to develop it to required level; 

i. Considering the complexity of African settings as acknowledge by the authors, i will advice authors should have focus on the implementation of the scheme in a Named Country as against Africa in general. This need made clear in both the title and the content. 

ii. It was difficult find succinct flow all through the presentation. I will advise authors to be guided by the journal guide regards structure of each section. 

iii. There was no mention of how secondary data used in the development of the manuscript was arrived at. This also need taking into consideration. 

iv. Both outcomes, discussion and recommendations need closer attention to ensure gaps discussed and advancement are logically linked for needed impact. 

v. Ensure that upto date and relevant materials are used to develop the work further 

Author Response

Response to Reviewer 1 Comments

Point 1: Considering the complexity of African settings as acknowledge by the authors, i will advice authors should have focus on the implementation of the scheme in a Named Country as against Africa in general. This need made clear in both the title and the content. 

Response 1:

Thank you for the suggestion.

Suggestion has been noted and incorporated. The title has been changed to “The Need for Nigeria to Embrace the Hygiene Rating Scheme”.

Point 2: It was difficult find succinct flow all through the presentation. I will advise authors to be guided by the journal guide regarding structure of each section. 

Response 2:

The manuscript has been revised to improve clarity and conciseness.

Point 3: There was no mention of how secondary data used in the development of the manuscript was arrived at. This also need taking into consideration. 

Response 3:

Secondary data used in this study was collected from peer reviewed journal articles, reports, conference papers, and internet articles. Key words such as *Hygiene rating scheme*, *foodborne disease*, *food safety*, and *Nigeria*, were entered in search engines such as “Google Scholar”, “Scopus”, “Web of Science”, and “PubMed”.

Point 4: Both outcomes, discussion and recommendations need closer attention to ensure gaps discussed and advancement are logically linked for needed impact.

Response 4: Discussion and recommendations sections have been written to make a better logical sense

Point 5: Ensure that upto date and relevant materials are used to develop the work further 

Response 5:

All new information in the revised manuscript is from recent and relevant material

Reviewer 2 Report

The authors in the acknowledgments thank the reviewers in advance for making insightful comments in improving the quality of the manuscript.
In my opinion this "thank you in advance", while very kind, is NOT logical.
The task of the reviewer, in my opinion, must be clearly distinguished from that of the Authors of an article or a review.
The reviewer's task is not to "enrich" the text with his authorial contributions (a fact that would make him an Author at this point).
The task of the reviewer, on the other hand, is to evaluate whether:
(1) the topic of the article/review is in line with the purposes of the journal,
(2) the authors have adequately developed the article in its various chapters,
(3) the Authors dealt correctly and effectively with all the topics they set out to touch.

Having said that, as my personal opinion, I would like to point out that:
(1) I can't quite understand if the Authors wanted to write an article or a mini-review, based,
(2) As set up, the article looks more like a practical contribution than a review, which I think is a valid approach,
(3) Being a practical contribution, I suggest to the Authors (if possible) to insert in the article one or two practical examples from which one can better understand how the evaluation system proposed by the Authors is applied.

In general, the contents of the article and the spirit of initiative that led the Authors to write it are appreciable.
I therefore believe that with the appropriate revisions, the article is in line with the contents required by the magazine.

I have NOT noticed any particular spelling errors in the text.

Author Response

Response to Reviewer 2 Comments

Point 1: I can't quite understand if the Authors wanted to write an article or a mini review, based

As set up, the article looks more like a practical contribution than a review, which I think is a valid approach,

Point 1: Being a practical contribution, I suggest to the Authors (if possible) insert in the article one or two practical examples from which one can better understand how the evaluation system proposed by the Authors is applied.

Response 3:

Adoption and Implementation of the Hygiene Rating Scheme in the UK

The UK government introduced the FHRS to improve food hygiene standards, promote transparency, and empower consumers to make informed choices about where to eat. The scheme was first launched in 2010 and has since been widely adopted nationwide [27]. 

The UK government worked closely with local authorities, food safety agencies, and industry stakeholders to develop and implement the HRS [28]. Extensive consultations were conducted to gather feedback and insights from various parties. This collaborative approach ensured the scheme was feasible, practical, and well-received by the industry [29]. The implementation of the scheme involved several key steps:

Standardized Inspection Criteria 

A set of criteria and guidelines were established to assess food establishments' hygiene practices [30]. These criteria cover food handling, cleanliness, and management procedures.

Inspection and Rating Process 

Trained environmental health officers to conduct routine inspections of food establishments. They assess compliance with the established criteria and assign a hygiene rating on a scale of 0 to 5. A rating of 5 indicates excellent hygiene practices, while 0 signifies urgent improvement is required [31].

Displaying Ratings 

Food businesses were mandated to display their hygiene ratings on their premises prominently. This enabled consumers to easily view and consider the hygiene ratings when making dining decisions [32].

Online Accessibility 

The hygiene ratings are also published on a dedicated website, allowing consumers to search and compare ratings for different establishments [33]. This online accessibility further enhances transparency and encourages businesses to maintain high hygiene standards.

The adoption of the HRS in the UK has yielded several significant impacts and benefits:

Improved Food Safety 

The scheme has driven food establishments to prioritize and enhance their hygiene practices. Businesses strive to achieve higher ratings, leading to improved food safety standards across the industry [34].

Consumer Empowerment 

The hygiene ratings empower consumers by providing them with easily accessible information about food establishments' hygiene standards. This transparency enables consumers to make informed choices and dines at higher-rated places, encouraging businesses to maintain good hygiene practices [33].

Industry Reputation and Compliance 

The scheme has positively influenced the overall reputation of the food service industry in the UK. Businesses with higher ratings are perceived as more trustworthy, leading to increased customer confidence. Moreover, the scheme has encouraged compliance as establishments strive to obtain better ratings [35].

Continuous Improvement 

The scheme operates as a dynamic process, prompting businesses to improve their hygiene practices continually. The regular inspections and public display of ratings create a feedback loop, motivating establishments to address any deficiencies and maintain high standards [36].

Adopting and implementing the HRS in the UK also faced certain challenges, including initial resistance from some food business owners, inconsistencies in interpretation by inspectors, and resource constraints for inspections [31]. However, ongoing efforts are made to address these challenges by providing support, training, and clear guidelines to inspectors and businesses.

Reviewer 3 Report

This manuscript presents the elements related to the need to implement the food rating scheme in Africa.

Although the notion of food safety is very important, the manuscript is quite general in presenting the importance and the impact of the scheme.

Clearly this is a huge issue that touches upon public service infrastructure, regulations, business operation, consumers' education, awareness and culture.

It is important for the authors to consider the message they want to give to the readers.

I feel the paper would greatly benefit from introducing a needs assessment proposal in each of the stakeholders involved and prioritize the activities in order to facilitate the adoption of the food rating scheme.

Another observation if the fact that the paper considers Africa as a whole whereas clearly several variations exist among the different countries that definitely hinder a horizontal approach. This should be clarified. Perhaps it would be more effective to use a single country in Africa (e.g. Nigeria) as an example to show how the scheme could be implemented.

Please provide a list of abbreviations and explain in text upon first use, example the following phrase:

"In many countries, processed foods are considered more hygienic due to monitoring and testing based on GMPs, GHPs, SSOPs, and HACCPs"

Author Response

Response to Reviewer 3 Comments

This manuscript presents the elements related to the need to implement the food rating scheme in Africa.

Although the notion of food safety is very important, the manuscript is quite general in presenting the importance and the impact of the scheme.

Clearly this is a huge issue that touches upon public service infrastructure, regulations, business operation, consumers' education, awareness and culture.

It is important for the authors to consider the message they want to give to the readers.

I feel the paper would greatly benefit from introducing a needs assessment proposal in each of the stakeholders involved and prioritize the activities in order to facilitate the adoption of the food rating scheme.

Another observation if the fact that the paper considers Africa as a whole whereas clearly several variations exist among the different countries that definitely hinder a horizontal approach. This should be clarified. Perhaps it would be more effective to use a single country in Africa (e.g. Nigeria) as an example to show how the scheme could be implemented.

Please provide a list of abbreviations and explain in text upon first use, example the following phrase:

"In many countries, processed foods are considered more hygienic due to monitoring and testing based on GMPs, GHPs, SSOPs, and HACCPs"

Response

Thank you for the suggestion.

The suggestion has been noted and incorporated. The title has been changed to “The Need for Nigeria to Embrace the Hygiene Rating Scheme”.

Also, all abbreviations used in the manuscript have been clearly defined

Involving stakeholders: Involving stakeholders such as government agencies, food service entrepreneurs, and consumer organizations in the scheme's development and implementation can help ensure its success.

Government Agencies and Local Authorities

To adopt the FHRS in Nigeria, NAFDAC must be actively involved. NAFDAC would have to collaborate with other private food safety agencies in Nigeria such as Rentokil Boecker to devise measures to create awareness of the scheme to food business operators, to conduct hygiene inspections, upload results of food business ratings to their website, and if possible, mandate food business operators to display result of ratings. They must provide trainings and support to ensure effective operation of the scheme. They must also be willing to take the right actions against food businesses with very low hygiene rating. The state and federal government must provide sufficient funding and resources for the smooth running of the scheme.

Food Business Operators

Food business operators must be willing to engage in hygiene inspections and comply with regulators to display results of ratings when mandatory.

Consumer

Once implemented, consumers must assess hygiene information about food business premises through FHRS certificates, and FSA websites.

Round 2

Reviewer 1 Report

Dear Authors 

Many thanks for time taken to improve the manuscript. 

I still find the method reported here is lacking in  content. You have now reported on the keyword words applied in various search engines to identify paper that inform the study outcome how ever there was no mention of number of materials generated, how these were screened and number taken forward that inform the paper. 

For guide, refer to PRISMA flow diagram to help improve this section further http://prisma-statement.org/prismastatement/flowdiagram.aspx?AspxAutoDetectCookieSupport=1 

Or 

Any similar pattern to help you improve on the work done.  

In addition,  i will recommend the use other related systematic review published material to guide you around reporting data related to the adopted materials 

I believe these alongside effort so far added to the manuscript, will help appeal to more wider audience  

Author Response

Response to Reviewer 1’s Comment

Point 1: I still find the method reported here is lacking in content. You have now reported on the keyword words applied in various search engines to identify paper that inform the study outcome however there was no mention of number of materials generated, how these were screened, and number taken forward that inform the paper.

Response 1:

Thank you for taking the time to review our article titled "The need for Nigeria to embrace the hygiene rating scheme." We appreciate your valuable feedback and the opportunity to address your concerns.

Regarding your comment on the lack of content in the methods section, we acknowledge the importance of transparency in research methodology, and we agree that it is crucial to provide a clear description of the methods used. To address this issue, we will revise the article to include the following information:

Several materials were generated during this study; however, only 57 were put forward for use. They comprise 29 recent journal articles, 14 recent studies published online, 1 textbook, and 13 reports, all relevant to the topic of discussion. All materials used were collected using search engines such as “Google Scholar”, “Scopus”, “Web of Science”, “PubMed”, and “Google”.

Screening process: The materials were screened based on some inclusion and exclusion criteria, such as relevance to the study objective and the quality and recency of the source material. However, some reports, such as the FAO (2005), were still used since they contained vital information and had not been revised.

Reviewer 2 Report

I have read the updated version of the article and I believe that in this version the requests for extension or synthesis that I requested in the first revision have been fulfilled. In my opinion, in the current version the article deserves to be published.

Author Response

Thank you

Reviewer 3 Report

The authors have improved the paper by addressing the comments and the text presents a much better proposal to improve food hygiene in Nigeria.

A minor suggestion: in line 342 reshape the paragraph named Consumer and merge it with lines 348-349 as follows .

"Consumers' Awareness-raising: Raising awareness about the importance of food safety and the benefits of the HRS among the public can help to encourage their support and engagement. Once implemented, consumers should assess hygiene information about food business premises through FHRS 343 certificates, and FSA websites. "

Author Response

Response to Reviewer 3’s Comment

A minor suggestion: in line 342 reshape the paragraph named Consumer and merge it with lines 348-349 as follows.

"Consumers' Awareness-raising: Raising awareness about the importance of food safety and the benefits of the HRS among the public can help to encourage their support and engagement. Once implemented, consumers should assess hygiene information about food business premises through FHRS 343 certificates, and FSA websites. "

Response

Your suggestion is well appreciated.

Suggestion has been implemented.

Consumers’ Awareness-raising: Raising awareness about the importance of food safety and the benefits of the HRS among the public can help to encourage their support and engagement. Once implemented, consumers should assess hygiene information about food business premises through FHRS 343 certificates, and FSA websites.